# Associations between Multimorbidity Patterns and Subsequent Labor Market Marginalization among Refugees and Swedish-Born Young Adults—A Nationwide Registered-Based Cohort Study

**DOI:** 10.3390/jpm11121305

**Published:** 2021-12-05

**Authors:** Jiaying Chen, Ellenor Mittendorfer-Rutz, Lisa Berg, Marie Norredam, Marit Sijbrandij, Peter Klimek

**Affiliations:** 1Section for Science of Complex Systems, CeMSIIS Medical University of Vienna, 1090 Vienna, Austria; jiaying.chen@meduniwien.ac.at; 2Complexity Science Hub Vienna, 1090 Vienna, Austria; 3Division of Insurance Medicine, Department of Clinical Neuroscience, Karolinska Institutet, 17177 Stockholm, Sweden; ellenor.mittendorfer-rutz@ki.se; 4Department of Public Health Sciences, Stockholm University, 10691 Stockholm, Sweden; lisa.berg@su.se; 5Centre for Health Equity Studies, Stockholm University/Karolinska Institutet, 10691 Stockholm, Sweden; 6Danish Research Centre for Migration, Ethnicity, and Health (MESU), Section for Health Services Research, Department of Public Health, University of Copenhagen, 1014 Copenhagen, Denmark; mano@sund.ku.dk; 7Section of Immigrant Medicine, Department of Infectious Diseases, University Hospital Hvidovre, 2650 Hvidovre, Denmark; 8Department of Clinical, Neuro- and Developmental Psychology and WHO Collaborating Centre for Research and Dissemination of Psychological Interventions, Vrije Universiteit, 1081 HV Amsterdam, The Netherlands; e.m.sijbrandij@vu.nl

**Keywords:** young adult, refugees, multimorbidity, unemployment, disability pensions, disease network

## Abstract

Background: Young refugees are at increased risk of labor market marginalization (LMM). We sought to examine whether the association of multimorbidity patterns and LMM differs in refugee youth compared to Swedish-born youth and identify the diagnostic groups driving this association. Methodology: We analyzed 249,245 individuals between 20–25 years, on 31 December 2011, from a combined Swedish registry. Refugees were matched 1:5 to Swedish-born youth. A multimorbidity score was computed from a network of disease co-occurrences in 2009–2011. LMM was defined as disability pension (DP) or >180 days of unemployment during 2012–2016. Relative risks (RR) of LMM were calculated for 114 diagnostic groups (2009–2011). The odds of LMM as a function of multimorbidity score were estimated using logistic regression. Results: 2841 (1.1%) individuals received DP and 16,323 (6.5%) experienced >180 annual days of unemployment during follow-up. Refugee youth had a marginally higher risk of DP (OR (95% CI): 1.59 (1.52, 1.67)) depending on their multimorbidity score compared to Swedish-born youth (OR (95% CI): 1.51 (1.48, 1.54)); no differences were found for unemployment (OR (95% CI): 1.15 (1.12, 1.17), 1.12 (1.10, 1.14), respectively). Diabetes mellitus and influenza/pneumonia elevated RR of DP in refugees (RRs (95% CI) 2.4 (1.02, 5.6) and 1.75 (0.88, 3.45), respectively); most diagnostic groups were associated with a higher risk for unemployment in refugees. Conclusion: Multimorbidity related similarly to LMM in refugees and Swedish-born youth, but different diagnoses drove these associations. Targeted prevention, screening, and early intervention strategies towards specific diagnoses may effectively reduce LMM in young adult refugees.

## 1. Introduction

The population of refugees has grown substantially over the last 15 years; 70.8 million people were forcibly displaced worldwide by the end of 2018; half of these refugees were under 18 years of age and 138,600 were unaccompanied children [1]. Adverse past and current migration experiences, including traumatic events and social exclusion in the host country, put young refugees at a higher risk for severe adverse health outcomes. These medical conditions could potentially affect their capabilities to integrate into the host country and establish themselves at the labor market.

Labor market marginalization (LMM) of the migrant population has become a major concern in European countries [2,3]. LMM is commonly defined as a state with temporary or permanent obstacles in labor market attachment [4]. LMM is often indicated by unemployment or other social insurance support that typically requires medical assessment, such as long-term sickness absence or a disability pension (DP) [2,4,5]. Young adults between 18 and 30 years of age can be granted temporary DP. The number of young adults granted disability pensions has increased in recent decades in Sweden [6], and the gap in unemployment rates between migrants and their native-born counterparts has persisted [7]. Moreover, younger age predicts increased vulnerability to unemployment, and economic disadvantage further increases this risk [3,8].

Mental disorders, along with their somatic comorbidities, are one of the main risk factors for LMM. For refugee youth in particular, increased life stress during migration can have a long-lasting negative impact on mental and physical health in adulthood [9,10,11,12] and thereby heightens their post-migration challenge. While previous studies have examined the relation between specific illnesses and risk of LMM, the effects of the co-occurrence of two or more diseases on subsequent LMM are inconsistent [13,14,15,16,17]. Diabetes mellitus and common mental disorders are frequent health problems in young adults and are known to increase the risk for other comorbid diseases as well, including heart diseases and other somatic disorders [13,17]. Given this multimorbidity in youth, the extent to which these co-occurring diseases are independent or combined risk factors for LMM is not clear, regardless of whether these risk factors differ between young refugees and the population of their host country.

Multimorbidity networks (here, a statistical term to describe the state of two or more diseases co-occurring in a patient) are crucial for understanding the biological mechanisms of the development of multiple diseases and their associations with each other [18,19]. The co-occurrence of diseases is the rule rather than exception, particularly in elderly patients, but also in youth [20]. While older age is associated with a higher number of chronic diseases, young adults are likely to have co-occurring mental illness and adverse physical health, particularly among economically deprived young adults [21]. The emerging field of network medicine has therefore embraced as its central tenet that diseases cannot be understood and treated in isolation from each other [22]. The aim of network medicine is to quantitatively elucidate how biological, social, and other types of network effects influence health to improve the identification, prevention, and treatment of diseases [23]. Such networks are made up of nodes (representing individual diseases) that are connected by links if the diseases tend to co-occur. Multimorbidity networks consist of age- and sex-specific clusters of diseases, i.e., groups of diagnoses that often co-occur with each other (e.g., the metabolic syndrome co-occurring with cardiovascular diseases; common mental disorders co-occurring with substance abuse) [20,24]. In addition, these networks often contain diagnoses that connect with many diagnoses from other clusters across the entire diagnostic spectrum; so called network hubs [25]. The hub diseases were shown to have positive association with mortality [18]. The appearance of multimorbidity network patterns—disease clusters and hubs connecting them—for young refugees is unknown, regardless of how their presence or absence impact labor market marginalization in refugee young adults. Here, we strive to increase our understanding of how multimorbidity patterns relate to economic inclusion and whether specific diagnostic groups show higher risks of subsequent LMM.

To the best of our knowledge, our study is the first to develop a concrete and practical way to leverage epidemiological analyses by personalized and patient-specific indicators derived from multimorbidity networks. Examining such disease patterns and their relation with subsequent LMM in young refugees is imperative to develop targeted and early intervention strategies to prevent long-term LMM. We sought to describe multimorbidity networks and identify specific diagnoses in these networks that are associated with LMM in refugee and Swedish-born young adults. The second aim was to identify the association between multimorbidity and LMM (unemployment and disability pensions) among young refugees in Sweden compared to Swedish-born individuals.

## 2. Materials and Methods

### 2.1. Study Design and Study Population

We used information from Swedish registries held by Statistics Sweden, the National Board of Health and Welfare, and the Social Insurance Agency to create a longitudinal prospective cohort study by linkage of de-identified data on an individual level. The study population included all refugees from 20–25 years of age (measured on 31 December 2011) residing in Sweden during the time interval from 1 January 2009 to 31 December 2011 (n = 42,721). Refugees were defined according to the following reasons of settlement: refugee status, through family reunification of refugees, in need of protection, or on humanitarian grounds. Each refugee was matched with five individuals in the comparison group based on age, sex, and type of living area (n = 213,605). Individuals in the comparison group were native-born Swedes with both parents born in Sweden. Among these 256,326 individuals, 5658 individuals were excluded because of an ongoing disability pension at baseline, and 1423 individuals were lost to the follow-up due to emigration at the beginning of the follow-up period. Therefore, 249,245 individuals, consisting of 41,516 refugee young adults and 207,729 Swedish-born young adults, were then followed for measures of labor market marginalization, from 1 January 2012 to 31 December 2016.

### 2.2. Data Sources

Information on sociodemographic, unemployment, and social benefits was obtained from Statistics Sweden, the Longitudinal Integration Database for Health Insurance and Labor Market Studies (LISA). The National Board of Health and Welfare provided information on date and diagnoses according to the International Classification of Diseases Tenth Edition (ICD-10) on inpatient and specialized outpatient health care in the “National Patient Register”. Data for sickness absences and disability pension (date and duration) was collected from Microdata for analysis of social security (MIDAS) derived from the Social Insurance Agency. Statistics Sweden’s longitudinal database for integration studies (STATIV) provided information on the reason for settlement in Sweden, e.g., refugee status.

### 2.3. Exposure Assessment

The exposures of interest were both the multimorbidity score, which was computed from the multimorbidity network, and the various diagnostic groups. The multimorbidity network was constructed from ICD-10-based disease co-occurrences observed between 1 January 2009 and 31 December 2011. A detailed description of the computation method is given below in the statistical analysis section. For the multimorbidity network, we examined associations of the ICD-10 diagnostic groups (main or secondary diagnoses) with LMM and evaluated the co-occurrence of diseases for each pair of groups (Appendix A, following a classification provided by the WHO). We included diagnostic groups containing ICD-10 codes from the ranges A00–N99, O00–O99, S00–T99, and V01–Y89 (Appendix A). One diagnostic group typically consisted of multiple diagnoses (e.g., the diagnostic group diabetes mellitus contained codes from the range E10–E14).

### 2.4. Outcome Ascertainment

The primary study outcome was LMM, which was assessed by disability pension (DP) and long-term unemployment, followed from 2012–2016. Disability pension was defined by whether an individual had been granted DP during follow-up period. Long term unemployment was measured as a binary variable, defined by whether the Swedish Public Employment Service reported more than 180 days of unemployment in a year.

### 2.5. Covariates

Demographic information was collected at baseline, including age, gender, family situation (married or cohabitant without children, married or cohabitant with children, single without children living at home, single with children living at home, youth younger than age of 20 years living at home), education level (0–9 years, 10–12 years, >12 years, missing), type of living area (big cities, medium-sized cities, small cities/villages), measures of whether an individual had a sickness absence spell of >90 net days in 2009–2011, and unemployment in 2009–2011. Missing information in education was defined as a separate category.

### 2.6. Statistical Analysis

#### 2.6.1. Multimorbidity Network

We examined the baseline characteristics of the complete cohort and groups by migration status (refugee or Swedish-born). Proportions between these two populations were compared using a two-tailed Z test. We computed the prevalence of each diagnostic group from 2009 to 2011. A network analysis was applied to identify multimorbidity patterns. Therefore, we computed all pairwise associations between individual ICD-10 diagnostic groups observed in 2009–2011 independent of migration status. The multimorbidity networks can be visualized as circles (nodes) connected by links (edges). Nodes in the network represent ICD-10 diagnostic groups, and node sizes indicate the number of individuals with the specific diagnoses. Links between two diagnostic groups indicate a statistical tendency to co-occur, as measured by their logarithmic odds ratio (the so-called “link strength”). To avoid finite-sized artefacts, we only included diagnostic groups with at least 100 cases and pairs that co-occurred at least 40 times. We used the disparity filter to address the multiple testing and class-imbalance problems, and a detailed description of the method was reported elsewhere [26]. In brief, the disparity filter corrects for statistical biases in the significance of link weights introduced by marginal frequencies that vary by several orders of magnitude.

#### 2.6.2. Logistic Regression

To assess the impact of multimorbidity on the outcome variables, we developed a novel personalized multimorbidity score. The score quantifies the LMM risk as a function of the diagnoses that this individual has given to the structure of the multimorbidity network. For an individual with a given disease *A*, we computed a network-induced outcome risk by summing the outcome difference over all significant links of node *A*, weighted by link strength (logarithmic odds ratio between diagnostic groups). The multimorbidity score of each individual was then the sum of these scores over all diseases of the patient. The score captures the total risk contributions to the outcome of all potential comorbidities of a patient’s diagnoses at baseline. Thus, the exposure information across the entire diagnostic spectrum can be collapsed into a single score in a personalized, patient-specific way. The advantage of our method is to lower the bias of an individual’s health condition due to the specification of selected diseases and to aid the identification of potential diseases leading to LMM. Crude and multivariate logistic regression was applied to assess the association of the multimorbidity score and LMM yielding odds ratios (OR) with 95% confidence intervals (CI). The differences of the association between refugees and Swedish-born youths were assessed by a two proportion Z test.

#### 2.6.3. Relative Risks of Diagnostic Groups

The bivariate association of each diagnostic group with the LMM was assessed. We identified the outcome difference between refugee and Swedish-born young adults in each diagnostic group by calculating the relative risk of LMM of young refugees in each diagnostic group, compared to matched Swedish-born young adults. The differences of relative risk between refugees and Swedish-born individuals were assessed by a Chi-square test or Fisher’s exact test in each corresponding diagnostic group. The color of nodes in the multimorbidity network represents the relative risk of LMM in refugee youths compared to Swedish-born youths. Diagnostic groups that related to lower relative risks in refugees are shown in green; higher relative risks are shown in purple; the higher intensity of the node color shows a higher risk of LMM in refugees or Swedish-born young adults. The analysis was performed using R 3.6.2 and SAS 9.4.

## 3. Results

### 3.1. Baseline Characteristics

Baseline characteristics are presented in Table 1. Mean age was 22.6 ± 1.71 years among the 249,245 young adults who met the inclusion criteria for the study. Refugees and Swedish-born individuals had similar age, sex, and type of living area due to the 1:5 matching. The refugee group had lower education compared to the Swedish-born: 24.6% vs. 8.49% received 0–9 years of education, respectively. The refugees had a lower frequency of previous sickness absence (1.12%, *p* < 0.0001) and higher previous unemployment (8.63 %, *p* < 0.0001) compared to Swedish-born with 1.70% and 2.34%, respectively. In total, 2841 individuals had been granted DP; 16,323 individuals had undergone long term unemployment during the follow-up period 2012–2016.

### 3.2. Multimorbidity Network

The multimorbidity networks for DP and unemployment are shown in Figure 1 and Figure 2, respectively. Several densely connected diagnostic groups also referred to as “clusters” were presented in the multimorbidity network: clusters of injuries, poisonings, and external causes of morbidity (chapters S, T, and V–X), pregnancy-related diagnoses (chapter O), respiratory diseases (J), and metabolic diseases (E). Mental and behavioral disorders formed a disease cluster, partly overlapping with another disease cluster of injuries and poisonings. Diagnoses related to pregnancy, childbirth, and the puerperium formed another cluster in proximity to the disease cluster of genitourinary diagnostic groups. Some diagnostic groups displayed as hubs by connecting different clusters of diagnostic groups. Here, episodic and paroxysmal disorders (G40–G47, which include sleep disorders) connected the mental health cluster with metabolic disorders (e.g., E70–E90) or disorders of the eye (groups from chapter H); pneumonia and influenza (J09–J18) connected respiratory diseases (chapter J) with other viral infections (chapter B) and several digestive disorders (chapter K).

Figure 1 showed that refugees had a lower risk of DP compared to Swedish-born for most diagnostic groups, as the multimorbidity network is dominated by green nodes. A few diagnoses showed an increased risk of DP for refugees, including diabetes mellitus (E10–E14) and influenza and pneumonia (J09–J18) with RRs (95% CI) of 2.4 (1.02, 5.6) and 1.75 (0.88, 3.45), respectively. The multimorbidity network depicted in Figure 2 was dominated by purple-colored nodes, suggesting that refugees had a higher risk of long-term unemployment in almost all diagnostic groups. External causes of morbidity and noninfective enteritis and colitis had the highest risk ratios for long-term unemployment in refugees with RRs (95% CI): 6.31 (4.45, 8.94) and 4.90 (3.44, 6.97), respectively.

Table 2 shows the associations of the multimorbidity score with LMM in refugees and Swedish-born young adults. The risk of LMM changes for each unit change in the multimorbidity score (standardized values with a mean of −0.012; 0.062 in the range [−0.6,16.9]; [−0.6,11.7] for unemployment in Swedish-born individuals and refugees, respectively; range [−0.5, 16.8], [−0.5,12.8] with a mean of 0.0012, −0.0063 for DP in Swedish-born individuals and refugees, respectively). The OR of DP was marginally higher in refugees (OR 1.59, 95% CI 1.52, 1.67) compared to Swedish-born individuals (OR 1.51, 95% CI 1.48, 1.54). The association between multimorbidity score and long-term unemployment did not differ between refugees (OR 1.15, 95% CI 1.12, 1.17) and Swedish-born individuals (OR 1.12, 95% CI 1.10, 1.14).

## 4. Discussion

In this study of 249,245 young adults in Sweden, more than 40,000 refugees were analyzed. The multimorbidity network showed that diabetes mellitus (E10–E14), influenza, and pneumonia (J09–J18) contributed higher risk of DP in refugee youths compared to the Swedish-born youths; external causes of morbidity (Y85–Y89), such as inflammatory bowel disease (K50–K52), showed higher risk of long-term unemployment in refugee youths, compared to the Swedish-born youths. We found a positive association between multimorbidity and LMM expressed as long-term unemployment and disability pension in refugees and Swedish-born young adults, with similar importance in both groups. However, results suggested that there are substantial differences in the way that cause individual diagnostic groups to contribute to that risk.

The multimorbidity network for young adult refugees and Swedish-born individuals consisted of well discernible clusters of co-occurring diseases, as well as diagnostic groups acting as hubs by connecting multiple clusters. For instance, a cluster of mental diagnoses (ICD chapter F) is clearly visible, meaning that, e.g., anxiety disorder (belonging to group ICD-10 F40–F49) was often comorbid with depression (F30–F39) and substance abuse (F10–F19). Previous literature showed that these highly connected diagnostic groups put patients at increased risk for diseases from large parts of the diagnostic spectrum, e.g., patients with sleep disorders frequently demonstrate metabolic, psychiatric, and other comorbidities [27]. Consequently, prevention efforts should be specifically targeted at such hubs in the multimorbidity network, in particular if they have a disproportionate number of connections to diagnostic groups with an increased risk for unemployment or DP.

In line with some previous findings [9], we found that refugees had lower relative risk estimates of DP and higher relative risk of unemployment in most of the specific diagnostic groups from the multimorbidity network, compared to Swedish-born youth. One explanation for these findings could be the lower overall employment rate in refugees. DP is often preceded by sickness absence, which in turn requires previous employment as an eligibility criterion. Moreover, refugees might be less knowledgeable of the social insurance system, which might contribute to lower chances of receiving DP.

The association between a higher relative risk of DP for young refugees related to some specific disorder groups, including diabetes mellitus (E10–E14), influenza, and pneumonia (J09–J18), may be explained by the higher severity of those diseases and other comorbidities. Severe influenza and pneumonia have been frequently associated with hospital readmission [28,29], residual lung damage [30], and other lower respiratory diseases [28,31], while the negative impact on the organ function may further lead to cognitive impairment and work capacity reduction [32]. The risk of readmission and in-adherence to treatment care could be higher in refugees, potentially resulting in a more severe health decline [29]. Here, migration experiences, such as traumatic life events, poor living conditions, and health behaviors during migration, may lead to more severe medical illness [33,34]; therefore, young refugees with these diseases might be more inclined to obtain DP. Additionally, previous studies have demonstrated that low health literacy in socially disadvantaged groups is related to adverse health outcomes and health disparities [35,36]. Moreover, refugees often experience barriers and limited access to healthcare delivery resources in their country of origin and during flight [2,34,37], which may lead to increasing severity of symptoms and urgency of receiving medical care after settlement [38,39]. Early prevention and health assessment regarding specific diagnostic groups, such as diabetes mellitus, influenza, and pneumonia, could potentially lower the health care burden and subsequent risk of DP in the host country for affected young refugees.

Several diagnostic groups were particularly associated with an increased risk of unemployment in refugees, such as external causes of morbidity (Y85–Y89) and inflammatory bowel disease (K50–K52). In general, multiple external factors may contribute to the pathway towards unemployment for young refugees, with one or several specific diagnostic groups from the multimorbidity network. First, low educational attainment may become a challenge for job seeking or advancement of young refugees [10,40,41]. In addition, cultural differences, long hour work conditions [39,42], and language barriers create obstacles for social inclusion, eventually contributing to unemployment in the host country. Moreover, some of the diagnostic groups associated with a high risk of unemployment in the multimorbidity network for refugees, such as external cause of injuries, diseases of the appendix, and inflammatory bowel disease, may require regular hospital visits, which in turn may affect the possibility to maintain a job [43,44,45], particularly if in precarious employment. These disorders might also be considerably more prevalent, of higher medical severity in young refugees, and have a strong effect on their occupational function and possibilities of obtaining and maintaining employment [34,43,44,45,46]. The high relative risk of unemployment in young refugees in some disorders warrants tailor-made interventions focusing on these disorders in order to lower the risk of unemployment for this vulnerable population. 

By quantifying multimorbidity by our novel multimorbidity score, our results demonstrate that multimorbidity is an important measure for unemployment and DP. To the best of our knowledge, this is the first study investigating the association between multimorbid and subsequent LMM. Previous studies have primarily examined specific diagnoses and their most relevant comorbid diseases with the risk of LMM, including neurodevelopmental disorder [16], mental disorder [47], and mild traumatic brain injury [14]. In our study, the individual multimorbidity score was computed from a disease co-occurrence network spanning the full diagnostic spectrum rather than pre-selecting specific diseases. It is therefore not straightforward to compare the findings of these studies with ours, as the characteristics of the study population, particularly regarding age range, as well as the choice of reference groups, differ. Our stratified analysis suggested that the pathways to LMM are quite similar in young adult refugees and Swedish-born individuals given their multimorbidity. The fact that the risk of LMM changed for each unit change in the multimorbidity score is of concern and warrants awareness regarding adequate assessment and treatment of comorbid disorders, as well as tight collaboration between medical disciplines and specializations in order to prevent early exit of the labor market.

In both refugees and Swedish-born young adults, we found that the multimorbidity score was more strongly associated with DP than unemployment. Granting of DP reflects a considerably higher medical severity of the underlying condition than unemployment and requires repeated medical assessments. It is therefore of outmost importance to include DP as a measure of LMM in related studies, as otherwise the extent of the adverse consequences of multimorbidity on subsequent occupational function and the chances of establishment at the labor market are seriously underestimated.

### Methodological Considerations

The strengths of our study include the large sample size and comprehensive health information with good quality and a long follow-up time, which provided high statistical power for the analyses. Moreover, the use of multimorbidity networks presents a novel methodology to assess the association of multimorbidity across the entire diagnostic spectrum, with regard to subsequent LMM in a personalized way. Our study also has limitations that are important to be mentioned. Diagnoses were collected from inpatient and specialized outpatient care, but information regarding the utilization of primary health care was lacking. This means that, primarily, diseases of higher medical severity were captured. For most diagnoses, we found rather low case numbers of refugees with DP; see Table 2. Further research is therefore needed to further corroborate our findings on relations between DP and specific diagnoses. Additionally, information regarding DP was unavailable from 2017 onward; however, the policy of DP remained unchanged throughout and after the study period. Lastly, as this study focusses on individuals in Sweden, the result may not be generalizable to other countries with different health care and social insurance systems.

## 5. Conclusions

Multimorbidity was associated with an elevated risk of LMM in young adult refugees and Swedish-born individuals, with only marginal differences. The multimorbidity network suggested that refugees had a lower relative risk of obtaining disability pension and a higher relative risk of long-term unemployment for most of the investigated diagnoses, compared to Swedish-born individuals with similar diagnoses. Early assessment and treatment of the identified diagnostic groups can potentially prevent early exit from the labor market for young refugees and Swedish-born youths.

## Figures and Tables

**Figure 1 jpm-11-01305-f001:**
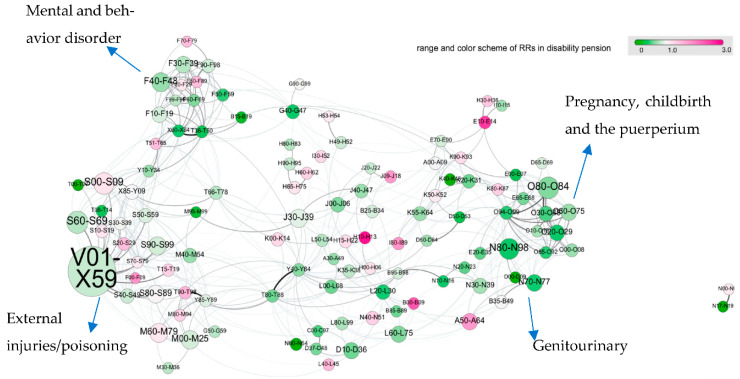
Multimorbidity network and disability pension for refugees versus Swedish-born young adults (n = 249,245). Each circle (node) corresponds to a group of ICD10 codes; the size of the circles is proportional to the number of patients in the cohort with that diagnoses; the color encodes the relative risk (RR) for DP between refugees and controls. The color intensity ranges from light green to bring purple. Purple (green) indicates that refugees (Swedish-born) with a diagnosis from the corresponding group have an increased risk for DP. The majority of diagnoses are therefore stronger risk factors for DP in Swedish-born young adults compared to refugees. Two diagnose groups are linked if they tend to co-occur in the same patients. The multimorbidity network has a clear cluster structure, where physiologically similar diseases (diagnoses from the same ICD chapter) form densely connected groups of nodes, which are bridged by certain diagnoses connecting diseases from different parts of the diagnostic spectrum; see text. The online version of the multimorbidity network can be found here: https://vis.csh.ac.at/remain/ (accessed on 24 November 2021). Label of each diagnostic group could be found in Appendix A.

**Figure 2 jpm-11-01305-f002:**
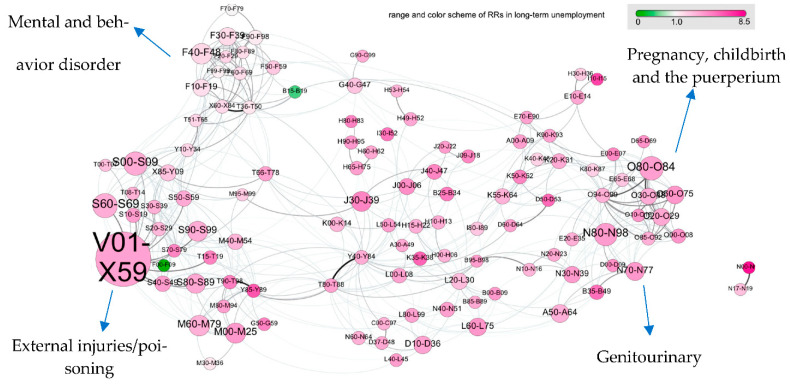
Multimorbidity network and unemployment for refugees versus Swedish-born young adults (n = 249,245). Each circle (node) corresponds to a group of ICD10 codes; the size of the circles is proportional to the number of patients in the cohort with that diagnoses; the color encodes the relative risk (RR) for unemployment between refugees and controls. The color intensity ranges from light green to bring purple. Purple (green) indicates that refugees (Swedish-born) with a diagnosis from the corresponding group have an increased risk for unemployment. The majority of diagnoses are therefore stronger risk factors for unemployment in refugees compared to Swedish-born individuals. Two diagnosed groups are linked if they tend to co-occur in the same patients. The multimorbidity network has a clear cluster structure, where physiologically similar diseases (diagnoses from the same ICD chapter) form densely connected groups of nodes, which are bridged by certain diagnoses connecting diseases from different parts of the diagnostic spectrum; see text. The online version of the multimorbidity network can be found here: https://vis.csh.ac.at/remain/ (accessed on 24 November 2021. The label of each diagnostic group can be found in Appendix A.

**Table 1 jpm-11-01305-t001:** Baseline characteristics of individuals aged 20–25 years of Swedish residents in 2011 (n = 249,245).

	Refugees	Swedish-Born ^1^	*p*-Value
	(n = 41,516)	(n = 207,729)	
Age (mean, sd)	22.6 (1.71)	22.6 (1.71)	
Gender (male, n(%))	22,189 (53.5)	110,982 (53.4)	
Family situation, n(%)			
Married or cohabitant without children	2127 (5.12)	2067 (1.00)	<0.0001
Married or cohabitant with children	4053 (9.76)	10,421 (5.02)	<0.0001
Single without children ^2^	30,130 (72.6)	170,272 (82.0)	<0.0001
Single with children	1267 (3.05)	2466 (1.19)	<0.0001
Youth (≤20 years) living at home	3939 (9.49)	22,503 (10.8)	<0.0001
Type of living area, n(%)			
Big cities	18,463 (44.5)	92,652 (44.6)	
Medium-sized cities	17,002 (41.0)	85,035 (40.9)	
Rural areas	6051 (14.6)	30,042 (14.5)	
Education levels, n(%)			
0–9 years	10,207 (24.6)	17,633 (8.49)	<0.0001
10–12 years	19,637 (47.3)	123,849 (59.6)	<0.0001
>12 years	9884 (23.8)	65,713 (31.6)	<0.0001
Missing	1788 (4.31)	534 (0.26)	
Previous Sickness absences, n(%) ^3^	467 (1.12)	3526 (1.70)	<0.0001
Previous Unemployment, n(%) ^4^	3583 (8.63)	4857 (2.34)	<0.0001

^1^ Swedish-born indicates the participants and that both his/her parents were native born Swedes; ^2^ Single with/without children living at home; ^3^ Sickness absence was defined as whether an individual received spells of >90 net days in 2009–2011; ^4^ Unemployment was defined as >180 days unemployment in 2009–2011.

**Table 2 jpm-11-01305-t002:** Odds ratios (95% confidence interval) regarding subsequent disability pension and long-term unemployment (2012–2016) according to the continuous multimorbidity score, stratified by refugee status in 249,245 young adults in Sweden.

	Refugees (n = 41,516)			Swedish-Born (n = 207,729)		
	n	Crude	Multivariate Model ^1^	n	Crude	Multivariate Model ^1^
Disability Pension as outcome						
Multimorbidity score in disability pension	379	1.67 (1.60, 1.75)	1.59 (1.52, 1.67)	2462	1.68 (1.66, 1.71)	1.51 (1.48, 1.54)
Unemployment as outcome						
Multimorbidity score in unemployment	7141	1.22 (1.19, 1.25)	1.15 (1.12, 1.17)	9182	1.26 (1.24, 1.28)	1.12 (1.10, 1.14)

^1^ Multivariable model adjusted for education, sickness absence during 2009–2011, long-term unemployment during 2009–2011.

## Data Availability

The data underlying the current study will be available upon reasonable request. Access to the dataset can be gained through application for register-based research at the Swedish Research Council.

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
