# Peer review of "Associations between Multimorbidity Patterns and Subsequent Labor Market Marginalization among Refugees and Swedish-Born Young Adults—A Nationwide Registered-Based Cohort Study"

_jpm, 2021, doi:10.3390/jpm11121305_

Round 1
Reviewer 1 Report
I thought the article was well done. The research questions posed were addressed through the network analysis completed by the authors. The article was well written and quite clear. I found the type of analysis and findings presented to be interesting and provided an innovative way to unpack these types of data. I do not have any recommendations for changes.Author Response
Dear reviewer,
Thank you so much for your time and comments on the manuscript! I truly appreciate your positive feedbacks.
Best,
Elise

Reviewer 2 Report
The study titled "Associations between multimorbidity patterns ... among refugees and Swedish-born young adults" represents a very well prepared manuscript, so I endorse it to be published as it is.
Author Response
Dear reviewer:
Thank you so much for your time and consideration on our manuscript! I really appreciate your positive comments!
Best,
Elise
